# The influence of cross-border mobility on the COVID-19 epidemic in Nordic countries

**Mikhail Shubin**[1]\*, **Hilde Kjelgaard Brustad**[2], **Jørgen Eriksson Midtbø**[3], **Felix Günther**[4], **Laura Alessandretti**[5], **Tapio Ala-Nissila**[6,7], **Gianpaolo Scalia Tomba**[4,8], **Mikko Kivelä**[9], **Louis Yat Hin Chan**[3], **Lasse Leskelä**[1]

**1** Department of Mathematics and Systems Analysis, Aalto University, Espoo, Finland, **2** Oslo Center for Biostatistics and Epidemiology, Oslo University Hospital, Oslo, Norway, **3** Department of Method Development and Analytics, Norwegian Institute of Public Health, Oslo, Norway, **4** Department of Mathematics, Stockholm University, Stockholm, Sweden, **5** DTU Compute, Technical University of Denmark, Copenhagen, Denmark, **6** Quantum Technology Finland Center of Excellence, Department of Applied Physics, Aalto University, Espoo, Finland, **7** Interdisciplinary Centre for Mathematical Modelling and Department of Mathematical Sciences, Loughborough University, Loughborough, United Kingdom, **8** Department of Mathematics, University of Rome Tor Vergata, Rome, Italy, **9** Department of Computer Science, Aalto University, Espoo, Finland

\* mikhail.shubin@aalto.fi

**Data Availability Statement:** All relevant data and code are published separately at https://gitlab.com/2pi360/covid_model_mobility_public.

## Abstract

Restrictions of cross-border mobility are typically used to prevent an emerging disease from entering a country in order to slow down its spread. However, such interventions can come with a significant societal cost and should thus be based on careful analysis and quantitative understanding on their effects. To this end, we model the influence of cross-border mobility on the spread of COVID-19 during 2020 in the neighbouring Nordic countries of Denmark, Finland, Norway and Sweden. We investigate the immediate impact of cross-border travel on disease spread and employ counterfactual scenarios to explore the cumulative effects of introducing additional infected individuals into a population during the ongoing epidemic. Our results indicate that the effect of inter-country mobility on epidemic growth is non-negligible essentially when there is sizeable mobility from a high prevalence country or countries to a low prevalence one. Our findings underscore the critical importance of accurate data and models on both epidemic progression and travel patterns in informing decisions related to inter-country mobility restrictions.

## Author summary

A typical intervention during pandemics such as COVID-19 is to restrict the mobility of individuals and thus prevent or slow down the spreading process. The role of within-country mobility has been studied in several countries, but the role of inter-country mobility is less well understood. To assess the effects of border closures that may cause significant economic and societal harm, it is necessary to understand in detail their efficacy from an epidemiological point of view. In the present work we model the effect of inter-country mobility on the spread of COVID-19 during 2020 in the neighbouring Nordic countries of Denmark, Finland, Norway and Sweden. We investigate the immediate

**Funding:** This work has been funded in part by the project 105572 NordicMathCovid as part of the Nordic Programme on Health and Welfare funded by NordForsk. The funder had no role in study design, data collection and analysis, decision to publish, or preparation of the manuscript.

**Competing interests:** The authors have declared that no competing interests exist.

impact of cross-border travel and employ counterfactual scenarios to explore the cumulative effects of introducing additional infected individuals into a population during the ongoing epidemic. Our results suggest that the effect of inter-country mobility on epidemic growth is non-negligible essentially when there is sizeable mobility from a high prevalence country or countries to a low prevalence one.

## 1 Introduction

Mobility and the consequent human contacts are the key factors for the spread of infectious diseases. Governments frequently impose and enforce border restrictions on individuals to prevent a disease from entering a country or to slow its spreading. However, such interventions can be problematic for economic, legal and social reasons. To assess the effects of border restrictions, quantitative modeling and prediction of their efficacy should be provided. This topic has experienced renewed interest each time a pandemic threat has emerged e.g., avian flu, SARS-CoV-1, Ebola and, most recently, COVID-19. Various approaches are possible, including descriptive studies [1], studies based on genomic analyses [2–4], and studies investigating social and health aspects [5]. However, the variety of mobility restriction schemes enacted by different countries during the COVID-19 pandemic and the general lack of quantitative evaluation of their differences and effects indicate a need for more analytical studies (e.g., [6, 7]). This is particularly important as restrictions on cross-border mobility may have serious detrimental effects on the economy and thus need to be proven to be effective on short and long time scales [8].

Most of the studies to date focus on the use of border restrictions in avoiding or delaying the start of a local epidemic. It has been shown (see [9] and references therein) that, if the restrictions are not 100% effective, they only postpone the start of the epidemic, and the delay time gained is usually short. Furthermore, restrictions must be enacted before the infection has been introduced in the country, since local spread dynamics will quickly dominate over importation. If infections are added in a later phase of the epidemic, the effect may be small because individuals that would be infected by the imported infection could be infected in any case due to the local epidemic [10].

Our interest in the present work is in the quantitative estimation of the effects of cross-border traffic on the number of infected in the Nordic countries sharing a common border during an ongoing epidemic. The COVID-19 pandemic in the Nordic countries during 2020 provides an interesting case study for the effect of border crossing traffic. First, the Nordic countries have a had a free movement agreement since 1954, but various exceptions occurred during the COVID-19 period. Second, there were major differences between the internal restrictions and the overall strategies carried out in each country, which led to large differences in the scale and timeline of the pandemic. While movement restrictions within these countries during the epidemic have been analysed, the analyses seem mostly to focus on legal or social aspects (see, e.g., [11, 12]). There are also analyses of the different intervention strategies and their effects in these countries, but without explicit analysis of the interactions between the countries [13].

The interaction between cross-border traffic and epidemic spread can be analysed in several ways, e.g., how epidemic outbreaks affect mobility (see, e.g., [6, 14, 15]) or vice versa, how mobility affects epidemic spread. For designing and enforcing effective but minimally disruptive restrictions on mobility and travel, the latter aspect is more important. Such studies can be both retrospective [10, 16], i.e., based on available data on traffic and disease, or prospective [17], i.e., entirely model-based. In both cases, however, the effects will usually be evaluated by

comparing scenarios with different amounts of traffic. In the retrospective case, this will involve counterfactual scenarios [13]. There are two main problems with such an approach. One is the need to specify how societies respond to new disease scenarios and the other is the choice of evaluation criteria of the differences between scenarios. Consider, for example, the term "first infective" (also sometimes referred as index case) from whom a local epidemic originates; this case is usually considered to be imported. Should we define the effect of importing this infection as the final size of the whole epidemic or just the individuals directly infected by that case? What if it were assumed that the infection would have been introduced anyway, even if the first infective had failed to infect anyone? It is thus important to clearly define measures of difference between scenarios and to distinguish the effects of internal and external forces of infection and of primary (direct) and secondary (consequential) effects of new infections into the population.

Our approach to evaluating the effects of inter-country mobility is to focus on two extreme cases: 1) *Primary effects* of mobility measured by descriptive statistics, i.e., the number of infected individuals that travel to and from each country. 2) *Secondary effects*, which we compute by running various counterfactual scenarios where we eliminate or restore cross-border traffic to pre-epidemic levels, but keep everything else unchanged, and simulate epidemic spreading under these new conditions. This is likely going to overestimate the number of infected over longer time intervals as large increases in the number of infections would probably have been met with stricter restrictions and changes in population behaviour. We thus present the effects over reasonable short forward time intervals. While neither of these evaluation methods give a realistic picture of the effects of mobility restrictions, they may serve as upper and lower bounds for the effects, and taken together they serve as a useful tool for assessing the range of possible outcomes. Our results indicate that the effect of inter-country mobility on epidemic growth is non-negligible essentially when there is sizeable mobility from a high prevalence country or countries to a low prevalence one.

To quantitatively assess the problem at hand we formulate an SIR mathematical model (representing the numbers of Susceptible, Infectious and Recovered individuals and their evolution over time in the Nordic countries; also known as an Eulerian approach) that explicitly uses estimates of cross-border movements of individuals. These movements are subdivided into short-term (commuter) and long-term visits in the receiving country. Combined with hospitalisation data from the modelled countries (and estimates of numbers of imported infections from the rest of the world), we infer the numbers of imported cases and their effects on within-country reproductive numbers during different phases of the disease spread. The model is then applied to the first year of the COVID-19 pandemic in the four Nordic countries sharing common borders: Denmark, Finland, Norway and Sweden. Traffic between the Nordic countries and the rest of the world is thus considered in the model.

The structure of this paper is as follows: In Section 2 we discuss the data and models used in the present study. Section 3 presents the measures and results for the primary effects, i.e., the direct impact of mobility. Section 4 presents the secondary effects of mobility, i.e., the counterfactual scenarios and their results. Finally, Section 5 summarises and discusses the conclusions of our study.

## 2 Data and models

Our study requires accurate and detailed data on the course of the epidemic within the countries, mobility between them, and a model that can be used to represent these data. Section 2.1 introduces the health data and Section 2.2 and S1 Appendix cover the mobility data. Section 2.3 describes the model (See also S2 Appendix for the derivation of the model).

## 2.1 Health data

To calibrate our model, we used the weekly numbers of new hospitalisations due to COVID-19. Data were obtained from the Finnish Institute for Health and Welfare (Finland), Socialstyrelsen [18] (Sweden), the Norwegian Institute of Public Health [19] (Denmark and Norway) and Our World in Data website [20] (rest of the world). We used aggregated data without age-based or regional stratification.

## 2.2 Mobility data

A central part of the research project has been the quantification of the number of individuals moving between the four bordering Nordic countries. To this end, we collected a data set consisting of 12 directed passenger flows between the Nordic countries Denmark (population 5.9 million people), Finland (5.5 M), Norway (5.4 M) and Sweden (10.5 M) and 4 directed passenger flows from the rest of the world into the 4 countries, and 4 directed passenger flows from the 4 countries to the rest of the world. The transmission model used to simulate the epidemic for each country has a daily temporal resolution. Hence, a daily temporal resolution is needed for the mobility data as well.

The directed flows represent the daily number of individuals travelling by air, road, railway and ferry. Fig 1A visualizes the temporally aggregated mobility between the countries in our data. We note that not all modes of transportation are relevant for all combinations of countries, e.g., no direct railway connection exists between Denmark and Norway. S1 Appendix presents more details on the collection of data for the different sources of transportation for the different countries.

We split the passenger flows into two categories: commuters and long-term travellers. We define commuters as one-day travellers, meaning individuals that exit a country and return to the same country the same day. The most probable transportation methods for commuters are by road. The road traffic data we have collected have an hourly temporal resolution. By modelling the road traffic, i.e., the time point of each border crossing, by a Gaussian mixture model (GMM), we identify components (Gaussian distributions) of the fitted model that correspond to commuting based on the estimated mean value (corresponding to a time point during the day) of the components. The commuting components then give the probability that a travel occurring at a specific time of the day is of commuter type. For more details, see S1 Appendix. Commuting also occurs by railway between Denmark and Sweden across the Öresund bridge. However, the temporal resolution of the railway data is not accurate enough to apply a GMM. We therefore assume the same fraction of commuters for railway traffic across the Öresund Bridge as for road traffic across it. Fig 1B visualizes how the mobility data are split into commuters and long-term travelers. The widths of the lines are proportional to the number of travels during the modelling period.

We collected data for the entire years of 2019 and 2020. The modelling period is the time interval 10 February 2020—31 December 2020. We used data for pre-pandemic year 2019 as a reference for the counterfactual scenarios to be discussed below.

## 2.3 Model

We use an extended Susceptible-Infected-Removed (SIR) model that tracks the epidemic status in each country. In our model time is discretised with a time step equal to one day. The model has three components: a deterministic *transmission model* simulates the epidemic, an *observation model* links the simulated number of infections to the observed data, and a *parameter model* defines the parameters for transmission and observation models. Fig 2 presents a graphical overview of the model while Table 1 summarizes the notation.

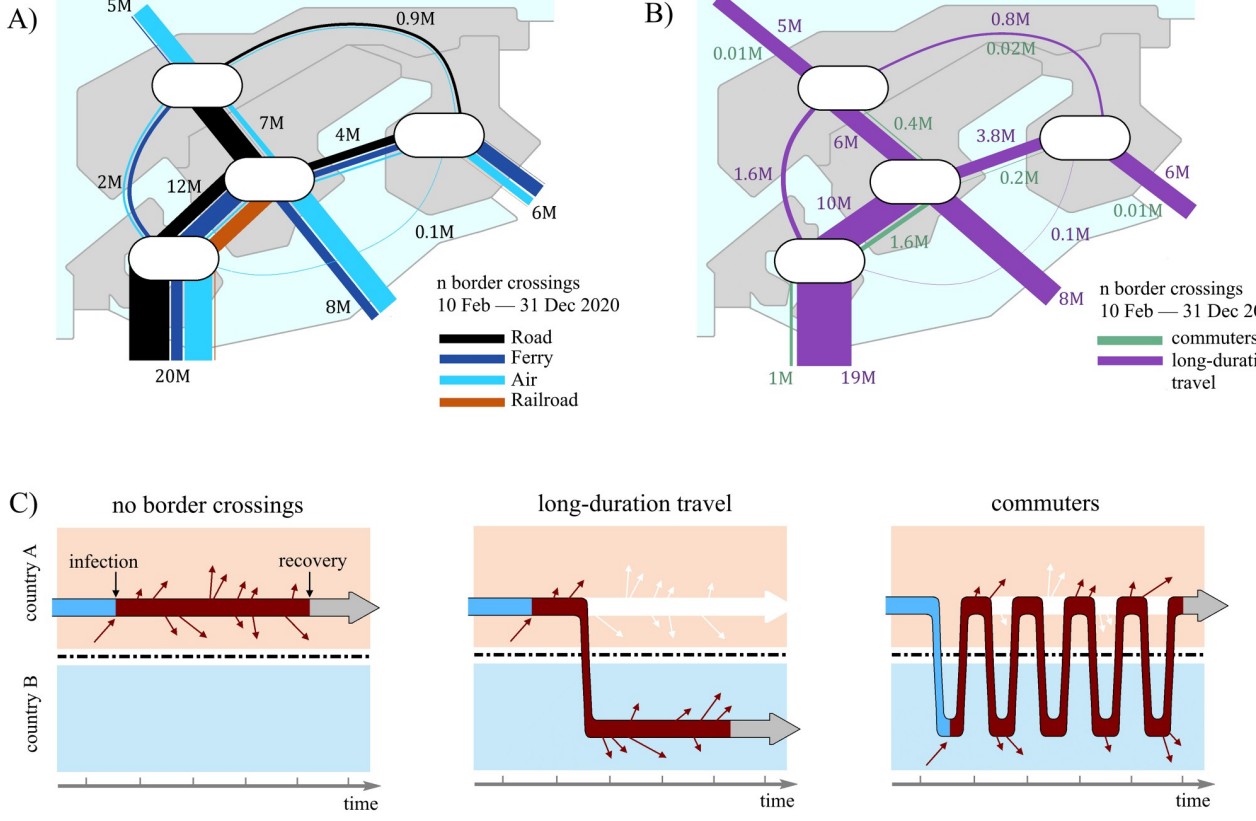

**Fig 1. Mobility between the four Nordic and the other countries, represented by lines connecting the origin and destination.** The widths of the lines are proportional to the numbers of border crossings during the modeling period (10 February—31 December 2020). (A): Line color represents the transportation type as indicated in the legend. Numbers near the lines show the total number of border crossings in millions (M). For comparison, populations of Denmark, Finland and Norway are about 5.5 M each, while that of Sweden is 10.5 M. (B) The number of commuters and long-term travellers (C) Schematic of commuters, long-term travellers and their representation in the model. Left sub-panel shows the base case scenario without any travel. Middle sub-panel shows an example of long-duration travel—an infected individual moves from country A to country B and stays there; this move creates extra infection pressure in B and reduces infection rate in country A. Right sub-panel shows an example of a commuter from country A getting infected in country B and continuing to move between countries.

**2.3.1 Transmission model.** The transmission model is spatially resolved at the level of each country and incorporates two types of mobilities: daily commuters and long-term travellers. We use an Eulerian approach where we keep track of individuals who are currently present in region $x$, and we follow the long-term flows of individuals across the regions [21]. In addition, we incorporate a Lagrangian model component to capture the effect of daily commuters as in [22]. We denote by $S_{t,x}$, $I_{t,x}$, and $R_{t,x}$ the expected numbers of susceptible, infectious, and removed individuals who are in the beginning of day $t \in \{0, 1, 2, \ldots\}$ located in country $x \in \{1, 2, 3, 4\}$. We allow population sizes of countries changes with time: $N_{t,x} = S_{t,x} + I_{t,x} + R_{t,x} \neq const$. Transmission characteristics are parameterised by the recovery rate $\gamma = 1/8$ (per day) and time-dependent country-specific reproduction rate $\mathcal{R}_{t,x} > 0$. Mobility is parameterised using external infectious flow counts $i_{t,x}^{\text{inflow}}$, commuter flow counts $D_{t,x,y}^{\text{com}}$, and long-term flow counts $D_{t,x,y}^{\text{long}}$ indexed by $t \geq 1$, and $x \neq y$.

The model evolves according to

$$S_{t+1,x} = \sum_y M_{t,x,y}^{\text{long}} S_{t,y} - i_{t,x}^{\text{new}}; \tag{1}$$

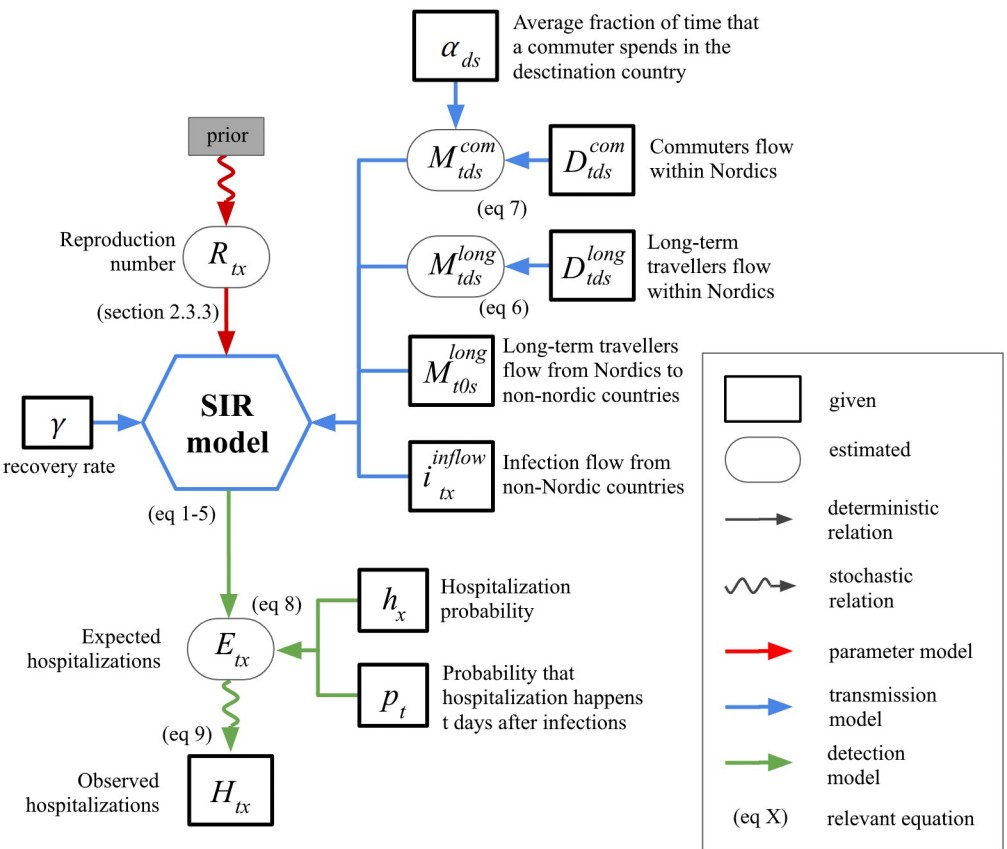

**Fig 2. Graphical representation of the SIR model adopted for the present study.** The model is spatially resolved at the level of each Nordic country and incorporates flows of travelers between them. The model part (transmission, observation, parameter) is indicated by the color. The schematic indicates both the model variables and data for hospitalisation and mobility. For more details, see Table 1 for notation and Section 2 on various components of the model.

$$I_{t+1,x} = \sum_{y} M_{t,x,y}^{\text{long}} I_{t,y} + i_{t,x}^{\text{new}} + i_{t,x}^{\text{inflow}} - I_{t,x} M_{t,o,x}^{\text{long}} - \gamma I_{t,x}; \tag{2}$$

$$R_{t+1,x} = \sum_{y} M_{t,x,y}^{\text{long}} R_{t,y} + \gamma I_{t,x}, \tag{3}$$

where

$$i_{t,x}^{\text{new}} = S_{t,x} \sum_{y} M_{t,y,x}^{\text{com}} r_{t,y} \tag{4}$$

is the number of residents of country $x$ infected on day $t$, and where

$$r_{t,y} = \mathcal{R}_{t,y} \gamma \sum_{z} M_{t,y,z}^{\text{com}} I_{t,z} / N_{t,y} \tag{5}$$

represents the risk of infection in country $y$ on day $t$. Here $M_{t,d,s}^{\text{com}}$ and $M_{t,d,s}^{\text{long}}$ are population-relative mobility matrices, defined as follows. Let $D_{t,d,s}^{\text{com}}$ and $D_{t,d,s}^{\text{long}}$ be the number of people from the

 

**Table 1. Notation glossary.**

| Parameter | Value | Definition |
|---|---|---|
| | | *Transmission model* |
| $\mathcal{R}_{t,x}$ | estimated | Time- and country-dependent reproduction number. This number aggregates the factors influencing the spread of infection: interventions, changes in population behaviour, seasonal effects and super-spreading outbreaks. |
| $S_{t,x}, I_{t,x}, R_{t,x}$ | estimated | Numbers of Susceptible, Infectious and Removed individuals per day per country |
| $N_{t,x}$ | $S_{t,x} + I_{t,x} + R_{t,x}$ | Population size |
| $\gamma$ | 1/8 (per day) | Recovery rate |
| $i_{t,x}^{\text{inflow}}$ | given | External infectious flow, i.e., number of infectious individuals arriving to country $x$ on day $t$ from outside of the Nordic countries |
| $i_{t,x}^{\text{new}}$ | Eq 4 | The number of residents of country $x$ infected on day $t$ |
| $r_{t,x}$ | Eq 5 | Risk of infection in country $x$ on day $t$, $r_{t,x} \in [0, 1]$ |
| $D_{t,d,s}^{\text{com}}, D_{t,d,s}^{\text{long}}$ | given | Counts of commuter and long-term travellers flow, i.e., the number of individuals going from country $s$ to $d$ on day $t$ |
| $M_{t,d,s}^{\text{com}}, M_{t,d,s}^{\text{long}}$ | Eqs 6 and 7 | Population-relative mobility matrices |
| $M_{t,o,s}^{\text{long}}$ | given | Population-relative outflow, i.e., the portion of population leaving country $s$ on the day $t$ to non-Nordic countries |
| $\alpha_{d,s}$ | 0.5 | Average fraction of time that a commuter from country $s$ spends in country $d$ during a daily trip |
| | | *Observation model* |
| $E_{t,x}$ | Eq 8 | Expected numbers of hospitalisations |
| $H_{t,x}$ | given | Observed number of hospitalisations |
| $h_x$ | 2% | The hospitalisation probability |
| $p_t$ | given | The probability that hospitalisation happens $t$ days after infection |

source country $s$ travelling to the destination country $d$ for a short (i.e., less than one day) or a long (i.e., one day or longer) visit, respectively. We define the relative matrices as:

$$M_{t,d,s}^{\text{long}} = \begin{cases} \dfrac{D_{t,d,s}^{\text{long}}}{N_{t,s}}, & \text{for} \quad s \neq d; \\ 1 - \dfrac{\sum_{x \neq s} D_{t,x,s}^{\text{long}}}{N_{t,s}}, & \text{for} \quad s = d; \end{cases} \tag{6}$$

$$M_{t,d,s}^{\text{com}} = \begin{cases} \alpha_{d,s} \dfrac{D_{t,d,s}^{\text{com}}}{N_{t,s}}, & \text{for} \quad s \neq d; \\ 1 - \alpha_{d,s} \dfrac{\sum_{x \neq s} D_{t,x,s}^{\text{com}}}{N_{t,s}}, & \text{for} \quad s = d, \end{cases} \tag{7}$$

where $\alpha_{d,s} = 0.5$ is a factor representing the average fraction of time that a commuter from country $s$ spends in country $d$ during a daily trip.

The model is initialized on day $t = 0$ corresponding to 10 February 2020, with 0.01% of the population being infected, i.e., such that

$$\begin{aligned} S_{0,x} &= 0.9999 N_x; \\ I_{0,x} &= 0.0001 N_x; \\ R_{0,x} &= 0, \end{aligned}$$

 

where $N_x$ is the population size of country $x$. Details of the derivation of the transmission model and its parameters are presented in S2 Appendix.

**2.3.2 Observation model.**　We define the expected number of hospitalisations on day $t$ in country $x$ as a convolution of the number of new infections:

$$E_{t,x} = \sum_{\tau=0}^{t-1} h_x p_{t-\tau} i_{t,x}^{\text{new}}, \tag{8}$$

where $h_x = 0.02$ is the hospitalisation probability in country $x$ and $p_d$ is the probability that hospitalisation happens $d$ days after infection. We set $p_d$ to be a negative binomial distribution with mean of 11 days and standard deviation of 5.

Let $H_{w,x}$ be the number of observed hospitalisations in country $x$ on week $w$. We link it to the expected numbers with the Negative Binomial (NB) distribution.

$$H_{w,x} \sim \text{NB}\left(\text{mean} = \sum_{t=7w}^{7w+6} E_{t,x}, \text{overdispersion} = 0.1\right). \tag{9}$$

**2.3.3 Parameter model and Bayesian parameter inference.**　The time-dependent reproduction numbers $\mathcal{R}_{t,x}$ are the only free parameters in our model. This parameter is supposed to aggregate the factors influencing the spread of infection: interventions, changes in population behaviour, seasonal effects and super-spreading outbreaks. Our goal is to fit the reproduction numbers $\mathcal{R}$ given the available data: number of hospitalisations $H$, inflow $i^{\text{inflow}}$ and mobility matrices $D^{\text{com}}$, $D^{\text{long}}$ and $M_{\bullet,0,\bullet}^{\text{long}}$. For each country, we estimate the posterior distribution

$$P(\mathcal{R}|H, D^{\text{com}}, D^{\text{long}}, i^{\text{inflow}}, M_{\bullet,0,\bullet}^{\text{long}}) \tag{10}$$

(where $\mathcal{R}$ represents a vector $\{\mathcal{R}_{t,x}, t \geq 0\}$) using an adaptive Markov chain Monte Carlo (MCMC) algorithm. We allow the chain to run for 15,000 warm-up iterations, and then we run it for 300,000 iterations recording every 10th. On each iteration $\mathcal{R}_x$ corresponding to each country is updated separately.

We construct the prior for $\mathcal{R}_x$ as a Gaussian random walk. The initial reproduction number is a priori sampled from a truncated normal distribution, limited to positive values only such that $\mathcal{R}_{t=0,x} \sim \text{Normal}_+(2, 0.5)$. The reproduction number on each subsequent Monday ($t > 0$, $t \bmod 7 = 0$) is sampled as $\mathcal{R}_{t,x} \sim \text{Normal}_+(\mathcal{R}_{t-7,x}, 0.25)$. The rest of the values are linearly interpolated between the closest Mondays. Proposal distributions for all batches of parameters are Multivariate Normal, with covariance adapted after each iteration. The details of the adaptation code are presented in [23] and the code implementing our model and fitting is available at gitlab.com/2pi360/covid_model_mobility_public.

# 3 Primary effects: Direct effects of imported and exported cases

## 3.1 Estimates of the effect of the mobility

This section defines the metrics we use to evaluate the effects of mobility. As the early data on the inflow of infections $i_{t,x}^{\text{inflow}}$ are unreliable, we only show these estimates for dates starting at April ($t = 51$).

**3.1.1 Net effect of mobility.** The right side of Eq 2 can be written as

$$\underbrace{I_{t,x}(1-\gamma)}_{\text{unrecovered existing infections}} + \underbrace{i_{t,x}^{\text{new}}}_{\substack{(A)\ \text{new infections among residents of } x}} + \underbrace{\sum_{y,y\neq x} M_{t,x,y}^{\text{long}} I_{t,y}}_{\substack{(B)\ \text{infected from other Nordics arriving to } x}}$$

$$-\underbrace{\sum_{y,y\neq x} M_{t,y,x}^{\text{long}} I_{t,x}}_{\substack{(C)\ \text{departing to other Nordic countries from } x}} + \underbrace{i_{t,x}^{\text{inflow}}}_{\substack{(D)\ \text{inflow from outside of Nordics to } x}} - \underbrace{M_{t,o,x}^{\text{long}} I_{t,x}}_{\substack{(E)\ \text{departing from } x \text{ to outside of Nordics}}}. \qquad (11)$$

Here the term $A$ refers to the new infections which occurred among the residents of country $x$, $B$ to infections coming to $x$ from other Nordic countries, $C$ to infections leaving from $x$ to other Nordic countries, $D$ to new infections coming to $x$ from non-Nordic countries, and $E$ to infections leaving from $x$ to non-Nordic countries. All the quantities $A$, $B$, $C$, $D$ and $E$ are positive. We refer to $Q_{t,x}^{\text{long term}} = B - C$ as the *net flow due to long-term travel* into $x$ and $Q_{t,x}^{\text{non-Nordic}} = D - E$ as the *net flow from non-Nordic countries* into $x$. The quantity

$$i_{t,x}^{\text{total new}} = i_{t,x}^{\text{new}} + Q_{t,x}^{\text{long term}} + Q_{t,x}^{\text{non-Nordic}} = A + B - C + D - E \qquad (12)$$

is referred to as the *total number of new infections*. Terms $B$ and $C$ contain contributions from the individual Nordic countries. Term $A$ defined in Eq 11 can be further written as:

$$i_{t,x}^{\text{new}} = \underbrace{\frac{\mathcal{R}_{t,x}\gamma}{N_{t,x}} S_{t,x} I_{t,x}}_{\substack{(F)\ \text{counterfactual local infections}}} - \underbrace{\frac{\mathcal{R}_{t,x}\gamma}{N_{t,x}} S_{t,x} I_{t,x}\left(1 - (M_{t,x,x}^{\text{com}})^2\right)}_{\substack{(G)\ \text{reduction in local infections due to commuters leaving}}} +$$

$$\underbrace{\frac{\mathcal{R}_{t,x}\gamma}{N_{t,x}} S_{t,x} M_{t,x,x}^{\text{com}} \sum_{y,y\neq x} M_{t,x,y}^{\text{com}} I_{t,x}}_{\substack{(H)\ \text{infections caused by commuters arriving from other Nordics to country } x}} + \underbrace{\sum_{y,y\neq x} \frac{\mathcal{R}_{t,y}\gamma}{N_{t,y}} S_{t,x} M_{t,y,x}^{\text{com}} \sum_{z} M_{t,y,z}^{\text{com}} I_{t,z}}_{\substack{(J)\ \text{commuters from country } x, \text{infected in other countries and returning back}}}. \qquad (13)$$

The quantities $F$, $G$, $H$ and $J$ are positive. Term $F$ refers to the counterfactual number of the local infections in country $x$ which would have occurred if no commuters had either arrived or left the country. Term $G$ refers to the number of infections inside country $x$ prevented by commuters leaving the country (this decreases both infection pressure and the pool of susceptibles). Term $H$ refers to the infections caused by arriving commuters in the local population. Finally, term $J$ represents the commuters from country $x$ who got infected while abroad and returned back to $x$. We refer to the term $i_{t,x}^{\text{local}} = F$ as *local infections* and to $Q_{t,x}^{\text{commuting}} = H + J - G$ as the *net commuting effect*. We further refer to the sum

$$Q_{t,x} = Q_{t,x}^{\text{long term}} + Q_{t,x}^{\text{non-Nordic}} + Q_{t,x}^{\text{commuters}} = B - C + D - E - G + H + J \qquad (14)$$

as the *net mobility effect*. In the following, we also define the *relative net mobility effect* as a net mobility effect divided by the total number of new infections, i.e.,

$$i_{t,x}^{\text{new rel}} = \frac{Q_{t,x}}{i_{t,x}^{\text{total new}}} = \frac{B - C + D - E - G + H + J}{A + B - C + D - E}. \qquad (15)$$

**3.1.2 Effective reproduction and multiplication numbers.** One of the main metrics of epidemic growth, the effective reproduction number $\mathcal{R}_{t,x}^{\text{eff}}$, is often defined as the average number of secondary infections caused by a single infected individual who got infected at day $t$ in country $x$. This definition describes local transmission and is not suitable for our purposes.

We thus suggest an *effective multiplication number* $\mathcal{M}_{t,x}^{\text{eff}}$ as the number of new infections that emerged in country $x$ for any reason during the average infectious period, divided by the number of new infections at day $t$.

Our definition of the parameter $\mathcal{R}$ already implicitly includes the effects from intervention, behavioral changes, weather etc., so the difference between $\mathcal{R}_t$ and $\mathcal{R}_t^{\text{eff}}$ can only be caused by reduction in the susceptible population. $\mathcal{R}_{t,x}^{\text{eff}}$ can be approximated as $\mathcal{R}_{t,x}S_{t,x}/N_{t,x}$, where $S_{t,x}/N_{t,x}$ is a remaining fraction of susceptibles. Alternatively, it can be approximated as a number of new infections per each infectious person on day $t$, multiplied by the average number of days one stays infectious $1/\gamma$. The number of new local infections in the absence of mobility is denoted as $i_{\tau,x}^{\text{total new}}$ (see Eq 13), and thus

$$\mathcal{R}_{t,x}^{\text{eff}} \approx \mathcal{R}_{t,x}\frac{S_{t,x}}{N_{t,x}} = \frac{i_{t,x}^{\text{local}}}{I_{t,x}\gamma}. \tag{16}$$

In analogy to the equation above, we approximate $\mathcal{M}_{t,x}$ by using the total number of new infections instead of only local ones:

$$\mathcal{M}_{t,x}^{\text{eff}} = \frac{i_{\tau,x}^{\text{total new}}}{I_{t,x}\gamma}. \tag{17}$$

Given that $\mathcal{R}$ is defined as a metric of epidemic growth due to the local transmission and $\mathcal{M}$ is a metric of growth due to all factors, $\mathcal{M} - \mathcal{R}$ represents the growth contributed by mobility. This kind of distinction between sources of epidemic growth has recently been made between the effects of local versus imported cases (see, e.g. [10, 24, 25]), although the exact definition of importation varies.

**3.1.3 Prevalence of infection.** The prevalence of local infections in country $x$ is defined as $\pi_{t,x} = I_{t,x}/N_x$; the prevalences of infection among commuters and among long-term travellers from Nordic countries, arriving to country $x$, are defined as

$$\pi_{t,x}^{\text{com}} = \frac{\sum_{y,y\neq x} I_{t,x}D_{t,x,y}^{\text{com}}/N_{t,y}}{\sum_{y,y\neq x} D_{t,x,y}^{\text{com}}}, \tag{18}$$

and

$$\pi_{t,x}^{\text{long}} = \frac{\sum_{y,y\neq x} I_{t,x}D_{t,x,y}^{\text{long}}/N_{t,y}}{\sum_{y,y\neq x} D_{t,x,y}^{\text{long}}}, \tag{19}$$

respectively. Prevalence of infection among long-term travellers from outside Nordic countries $\pi_{t,x}^{\text{inflow}}$ computed directly from the data, see S2 Appendix for details.

## 3.2 Results

We will next describe the results of applying our approach to the four neighboring Nordic countries during April–December 2020. The progress of the epidemic differs between the countries (see Fig 3, also S3 Appendix Fig F). Denmark, Finland and Norway initially followed the same epidemic trajectory: in early April the incidence rate, which we always give as new infections per day per 10,000 inhabitants, was about 5. This incidence rate quickly decreased and stayed below $1 - 2$ until October. On the other hand, Sweden had an incidence rate of 20 during the whole of April and this rate dropped to 1 only by July. In October–December Finland, Norway and Sweden experienced a second wave which reached incidence rates of 3 for

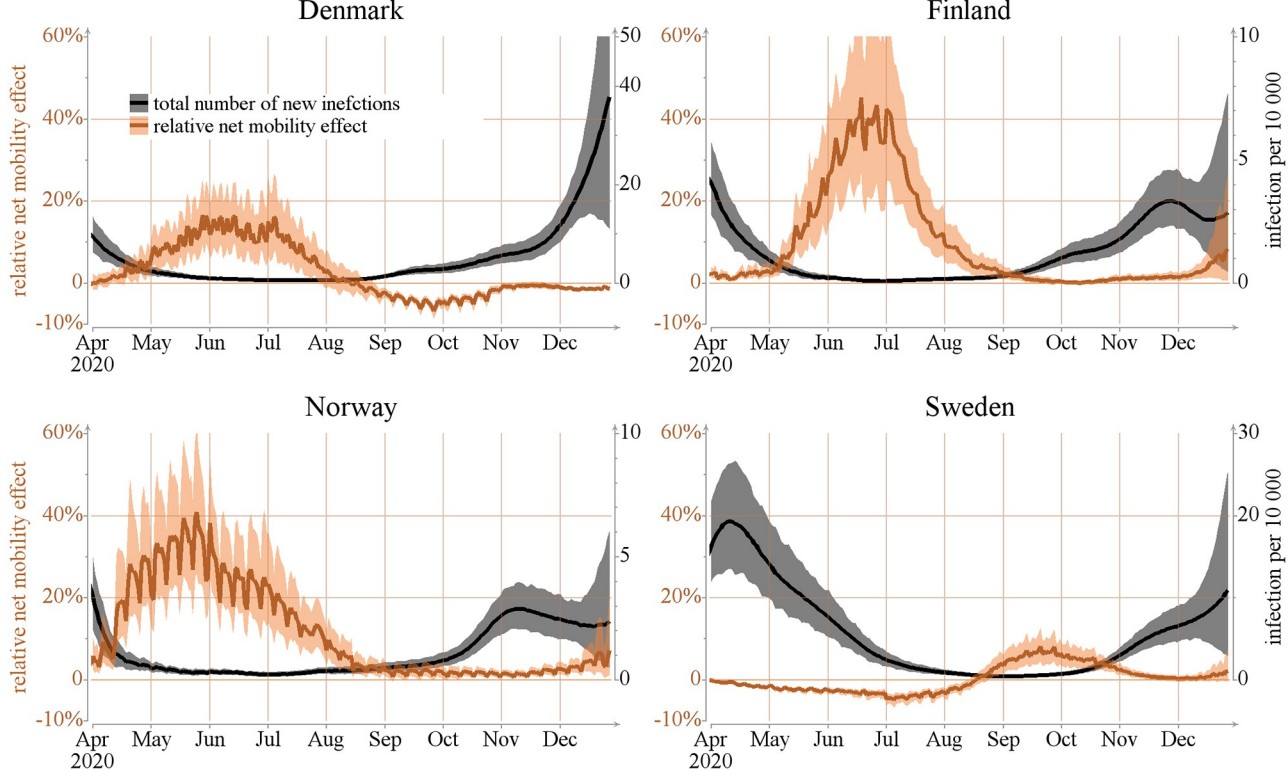

**Fig 3.** Black color: estimates of the total number of new infections $i_{t,x}^{\text{total new}}$ of Eq 13 per day per 10,000 individuals (lines show posterior mean and colored areas show 90% posterior intervals). Orange: estimates of the relative net mobility effect of Eq 15 $i_{t,x}^{\text{new rel}}$, defined by dividing the net mobility effect (cf. Eq 14) by the total number of new infections (lines show posterior mean and colored areas show 90% posterior intervals). Note that the net mobility effect can be negative, meaning that the country is a net exporter of infections.

Finland and Norway and 10 for Sweden. In Denmark, the second wave started earlier, in September and reached an incidence rate of 40 by December.

The first interesting result from our modeling study is that during the time period considered here, the net effect of inter-country mobility is very small as compared to the number of local infections. To quantify its effect, we computed the *relative net mobility effect* $i_{t,x}^{\text{new rel}} = Q_{t,x}/i_{t,x}^{\text{total new}}$ (see Eq 15). As seen in Fig 3, this fraction is close to zero in Finland almost until mid-May. The relative net mobility effect then peaks around June–July, reaching 40% (0.04 out of the total 0.1 incidence rate is explained by mobility). Norway exhibits a similar pattern to Finland, with the peak time starting earlier already in April, stretching slightly longer in time and reaching 30% (0.1 out of the total 0.3 incidence rate is explained by mobility). Denmark displays a similar, but even less pronounced, peak during spring and summer, reaching 4% (0.002 out of the total 0.05 incidence rate is explained by mobility). However, Denmark becomes a net exporter of infections in August, which means that there were more infected individuals travelling out of the country than coming in during this time period. Sweden on the other hand is a net exporter of infections in our model until late August, however $i_{t,x}^{\text{new rel}}$ never exceeds 10%.

Our model shows that for the inter-country mobility to have a non-negligible effect, three conditions have to be met: (i) the number of new infections in the target country must be very low, (ii) the number of infections in the source country must be large compared to the target

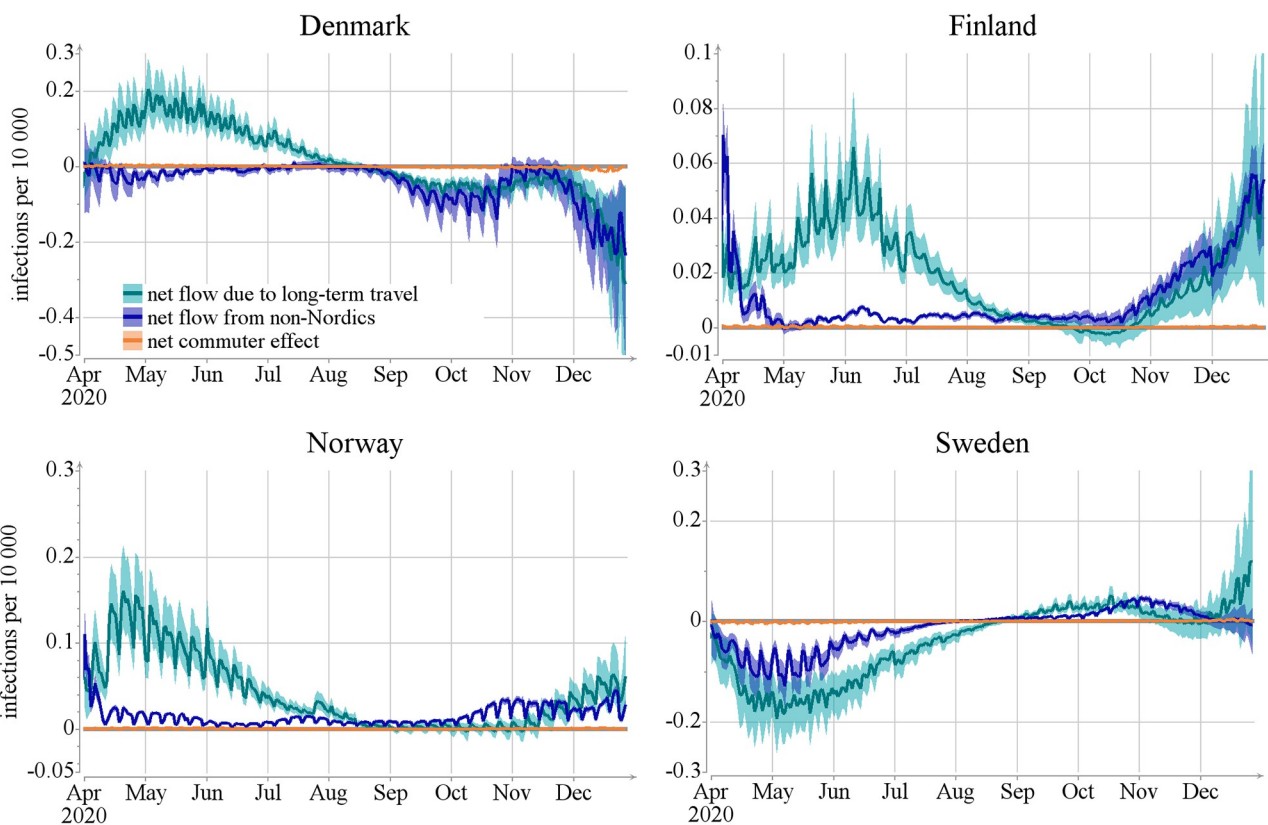

**Fig 4. Results of the net mobility effect on the number of infections per day per 10,000 caused by different modes of travel: $Q_{t,x}^{\text{long term}}$, $Q_{t,x}^{\text{non-Nordic}}$ and $Q_{t,x}^{\text{commuters}}$ (see text for details).** Lines show posterior mean and colored areas show 90% posterior intervals.

country, and (iii) the mobility between the countries must be sufficiently large to transfer sizable amounts of infection. There are two time periods when the first two conditions were met, namely April–August 2020 when Sweden had significantly more new infections than the other countries, and December 2020 when Denmark had a high incidence (see Fig 3). While difference in incidence between two countries is necessary for border crossings to make a difference in our model, it is not a sufficient condition, because the rate of border crossings might not be large enough to enhance the rate of infections. This is the case in the two aforementioned instances, as will be discussed below.

We next turn to the question of what type of travellers brought in the infections, and use the division of border crossings into three groups described in Section 2.2: commuters, long-term travellers from the four Nordic countries, and travellers from the rest of the world. Fig 4 shows the effect of the estimated total net flow of infected travellers on the number of infections per day per 10,000 using this decomposition. The contribution of individuals identified as commuters is consistently very low in our model. This is due to a combination of two effects: the relatively low number of commuters and the fact that the model yields a smaller effect per border crossing for commuters because they spend only part of the day in the country whereas long-term travellers are assumed to reside in the country for their remaining infectious period. See S3 Appendix Fig G for the relative numbers and Fig H for the further split of flows.

The data on incidences in Fig 3 and on the traveller types of Fig 4 suggest that the significant increase in the relative number of imported infections during May—August 2020 in

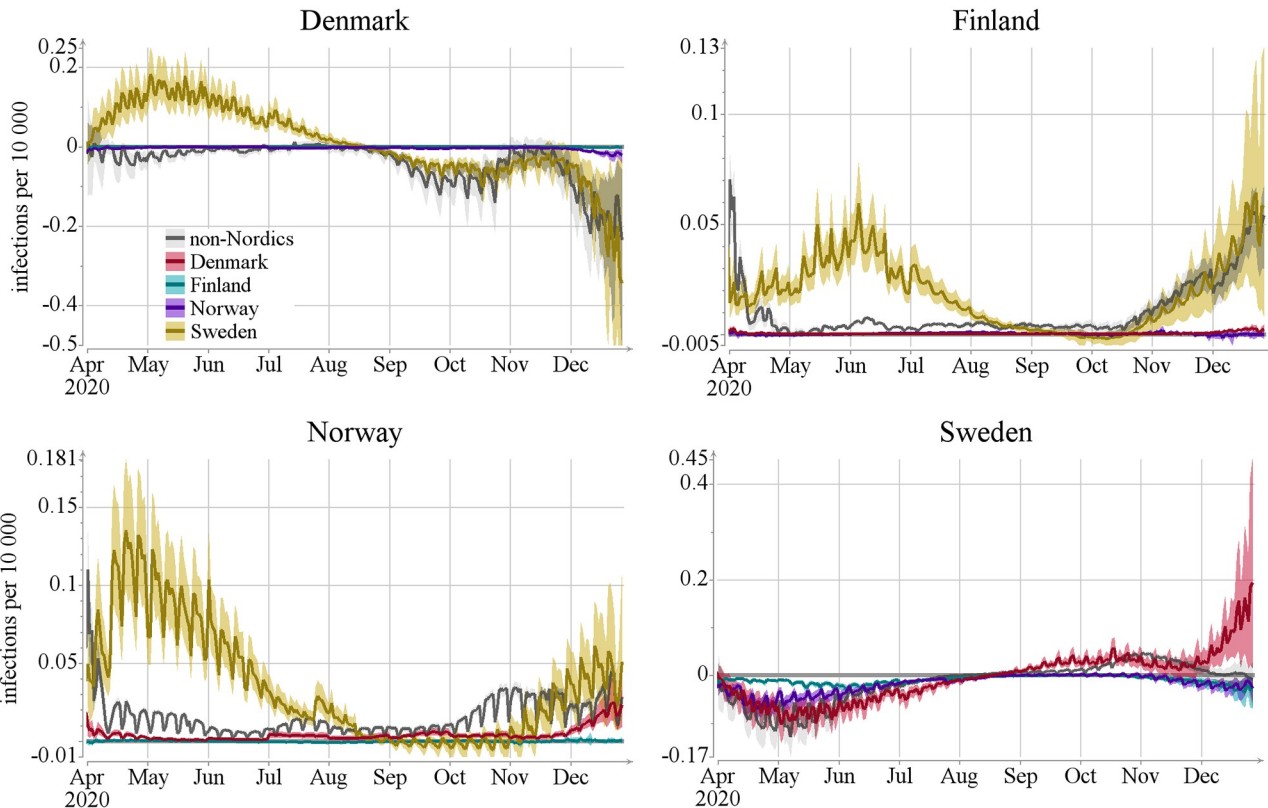

**Fig 5. Daily net mobility effect on the number of infections per day per 10,000 individuals from each source country.** The color legends are in the upper right panel. The lines show posterior mean and colored areas show 90% posterior intervals.

Finland, Norway, and Denmark was due to infected travelers from Sweden. This is confirmed by Fig 5, which exhibits the effect of the countrywise net flows on the number of infections per day per 10,000 (refer to S3 Appendix Fig I for relative numbers). Note, however, that in December 2020, and to some extent already starting from September, Denmark became a net contributor of infections to the other Nordic countries.

The results shown here carry somewhat contradictory messages. One one hand, they show that the net effect of the inter-country mobility is very low. On the other hand, when local transmission rate is low, incoming infections may play a substantial part in the epidemic. To clarify the message, we can examine the epidemic trajectories from another angle, namely by comparing the effective reproduction and multiplication numbers $\mathcal{R}^{\text{eff}}$ and $\mathcal{M}^{\text{eff}}$, respectively (Fig 6). When $\mathcal{M}^{\text{eff}} > \mathcal{R}^{\text{eff}}$ mobility introduces extra infections into the population, but when $\mathcal{M}^{\text{eff}} < \mathcal{R}^{\text{eff}}$ mobility removes infections from the population. In particular, when $\mathcal{M}^{\text{eff}} > 1$ and $\mathcal{R}^{\text{eff}} < 1$ the epidemic can only grow because of the inter-country mobility. Such a situation occurred for Finland and Norway during June—August 2020.

While the analysis above tells us about the importance of the border crossings for the epidemic pressure on the Nordic countries, it does not directly tell us about the effectiveness of interventions targeting traffic across borders. Here, instead of asking the question whether of not interventions should be implemented, we want to answer the question of whether it would be more effective to enforce interventions on the local population or on the population crossing the borders. To this end, we compute the prevalence, i.e., the proportion of infectious

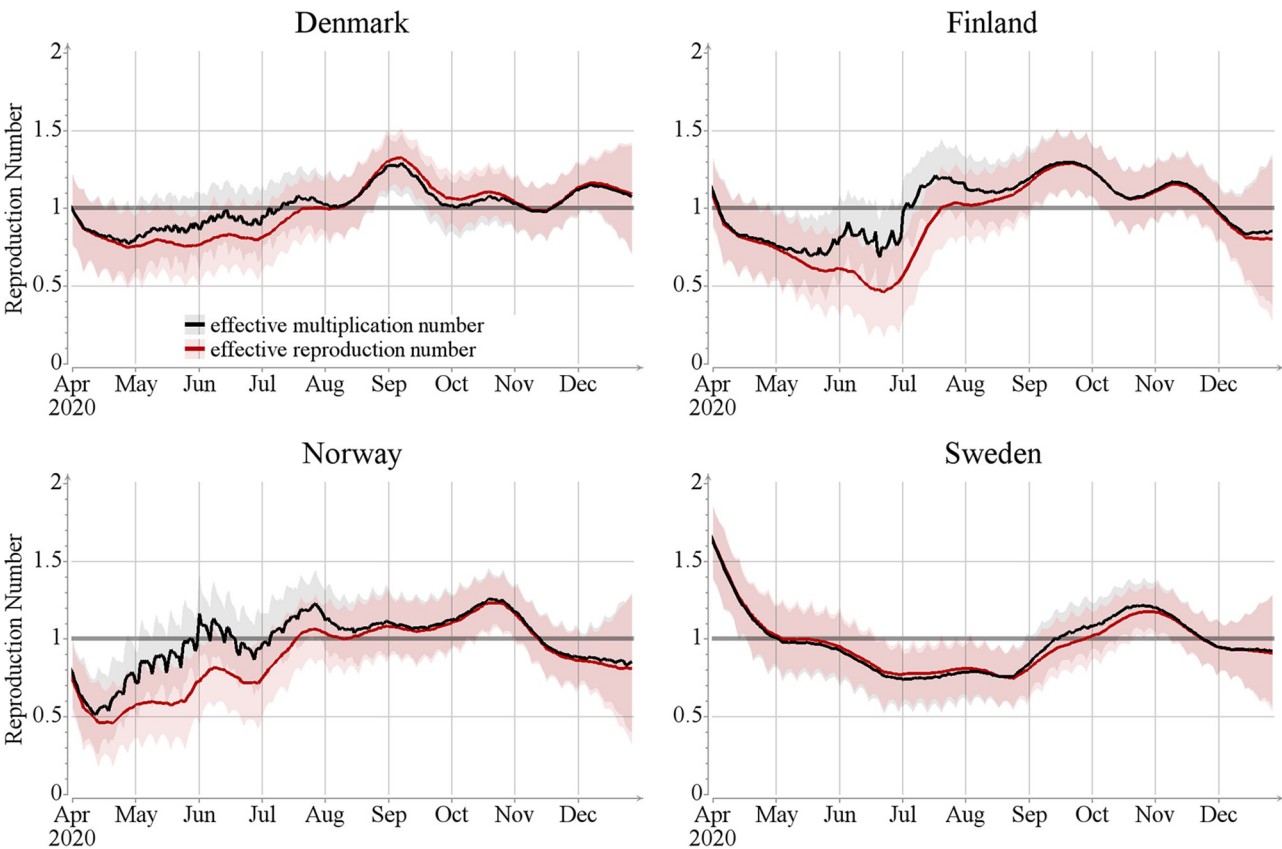

**Fig 6. Estimates of daily effective reproduction number $\mathcal{R}^{\text{eff}}_{t,x}$ and multiplication number $\mathcal{M}^{\text{eff}}_{t,x}$, as approximated in Eqs 16 and 17.** The lines show posterior mean and colored areas show 90% posterior intervals.

individuals, among the local population ($I_a/N_a$) and the long-term travellers arriving to the country from Nordics and non-Nordic countries (see Fig 7). There are very large differences in the prevalences of local and travelling populations. Most notably, in Norway and Finland the Nordic travellers are more than one hundred times more likely to be infectious than the local population during the May—August of 2020. This indicates that every test done at the border during that period (for individuals without symptoms and no knowledge of exposure) was potentially more than a hundred times more efficient than tests for the local populations. Similarly, offsetting the effect of a single border crossing could potentially require much larger local movement restrictions. The opposite is true for Sweden, where until mid August, the prevalence of the local population is larger than the prevalence of Nordic travellers.

## 4 Secondary effects: Counterfactual scenarios

The previous section focused on the number of infections coming into a given country. However, such numbers might underestimate the impact of border crossings: imported infections can lead to further infections, which are then classified as local infections as they are taking place within the country. These further infections could have been (partly) prevented if the original case in the infection chain were prevented. We use counterfactual scenarios to investigate the effect of full infection chains caused by border traffic. The border effect is isolated by

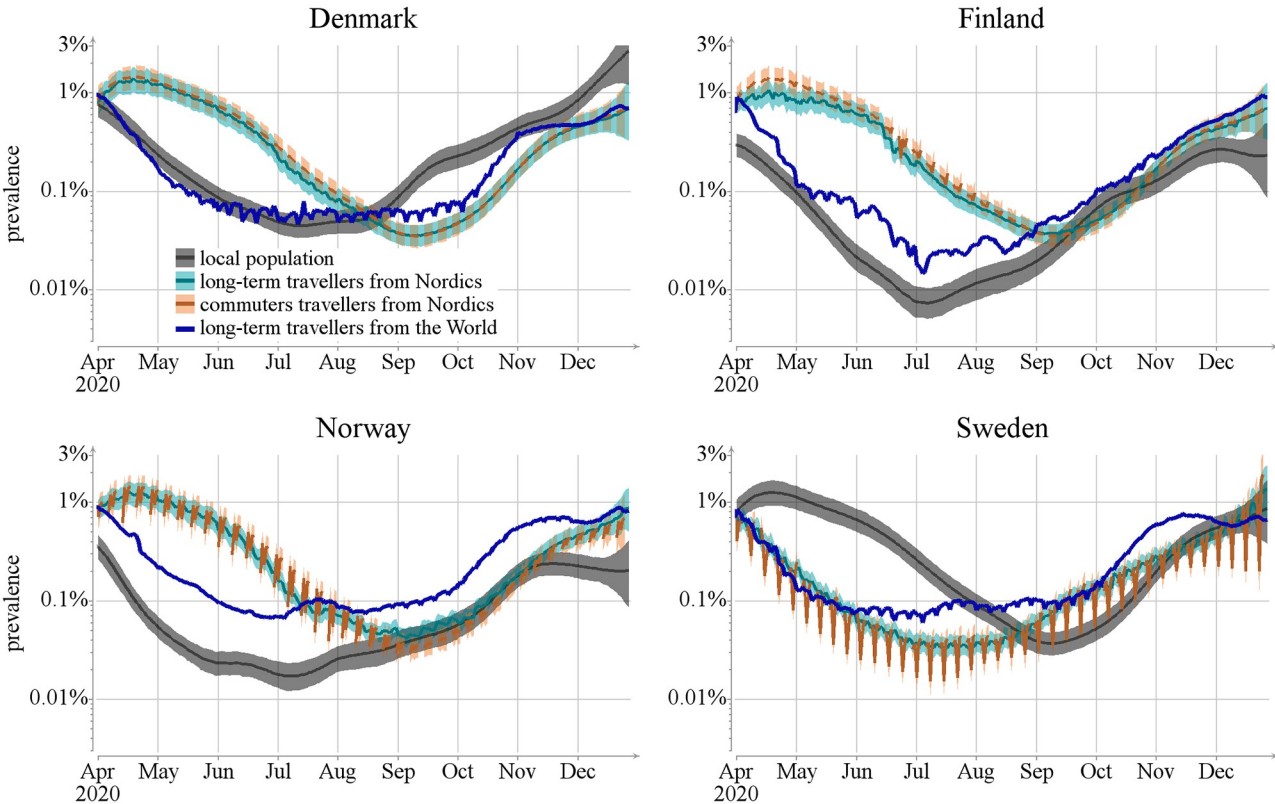

**Fig 7. Prevalence among the local population for different travel types, and for long-term travellers visiting a country.** Lines show posterior mean and colored areas show 90% posterior intervals.

changing the rate of mobility while keeping everything else in the model (including the disease spreading parameters) constant.

The counterfactual scenarios are not necessarily accurate predictions of what would have happened if travel restrictions or testing policies at the border were changed. There are two main reasons: firstly, changing policies at borders can alter the behaviour within the country in multiple ways. Some people may take border closure as a signal from authorities of the seriousness of the epidemic and abstain from all types of contacts. Others might instead choose to travel more within the country. Secondly, the within-country policies and behaviour are affected by the current epidemic situation, but in our counterfactual scenarios we assume that the country-specific reproduction rates $\mathcal{R}_{t,x}$ remain unchanged. For example, if the overall disease burden went down due to restricted cross-border traffic, this could imply that people mix more inside a country and thereby increase the reproduction rate. These responses are difficult to model due to the general nonlinearities and lack of monotonicity inherent in the transmission dynamics [26].

Since we do not model the societal responses to the current epidemic situation, we expect the reliability of our estimates to go down for longer time horizons. Nevertheless, our estimates should serve as an upper bound for the change in number of infected individuals due to border traffic when this traffic increases the number of infections, because more infections would likely result in a reaction decreasing the local reproduction rates rather than increasing them. Similarly, if the changes in border traffic caused a lower number of infections within the country, the counterfactual would serve as a lower bound for the effect as lower number of infected

people would likely lead to unchanged or higher reproduction numbers. Also, we do not investigate realistic scenarios where one would slightly adjust the numbers of travellers in an adaptive manner. Thus, short-term estimates should indicate the right order of magnitude and direction of the effect of counterfactual scenarios.

## 4.1 Defining the counterfactuals

We focus on the two extreme cases to find the upper and lower limits of the effects of cross-border traffic: We model what-would-have-been cases as two scenarios, where either all mobility is cancelled (zero mobility matrices) or where the mobility is restored to the 2019 pre-pandemic level. We compare these scenarios to a baseline scenario which is the model fitted to the real data from the year 2020 as described in Section 2.3.

We use the same sample of reproduction numbers, $S_{\mathcal{R}}$, sampled from the posterior distribution of the model fitted to the mobility (and health data) from year 2020 in all of our scenarios. For each of these samples $\mathcal{R} \in S_{\mathcal{R}}$, we compute the number of infected individuals at each time step $t$ given the reproduction number and mobility data $I_{t,x}(\mathcal{R}, D^{\mathrm{com}}, D^{\mathrm{long}}, i^{\mathrm{inflow}}, M^{\mathrm{long}}_{\bullet,0,\bullet})$. Note that this computation is deterministic in our SIR model. We then estimate the posterior mean numbers of infected individuals,

$$E(I_{t,x}) \approx \frac{1}{|S_{\mathcal{R}}|} \sum I_{t,x}(\mathcal{R}, D^{\mathrm{com}}, D^{\mathrm{long}}, i^{\mathrm{inflow}}, M^{\mathrm{long}}_{\bullet,o,\bullet}) . \tag{20}$$

We do all of our computations using $|S_{\mathcal{R}}| = 1000$ samples.

To construct a counterfactual model, we substitute the appropriate values of the mobility matrices $D^{\mathrm{short}}$, $D^{\mathrm{long}}$, $i^{\mathrm{inflow}}$, and $M^{\mathrm{long}}_{\bullet,o,\bullet}$. To express the scenario for the border closure starting on day $t$, we modify the mobility data by filling the matrices corresponding to day $t$ and later with zeros. For example, to model the effect of a border closure starting in May 2020, we set all values in the matrices starting from 1 May 2020 to zero. For the scenario of border reopening, the mobility matrices at time $t$ and after are filled with data from 2019. We then recompute the expected number of infected individuals $E(I)$ in Eq (20) with the modified matrices but the same sample of the reproduction numbers $\mathcal{R}$.

We investigate the timing of the hypothetical interventions by varying the starting time at the first day of each month. We show the resulting counterfactual trajectories of $I$ only for 50 days after the start of the scenario, assuming that after 50 days, *ceteris paribus*, assumptions for $\mathcal{R}$ become unrealistic.

## 4.2 Results

For completely closing all border traffic see Fig 8, and returning back to the traffic patterns of 2019 see Fig 9 (see also S3 Appendix Fig J and K). In Finland and Norway, removing the border traffic in May and June leads to a large reduction in the number of infections during the fifty-day period in our model. Removing the border traffic in July still has a significant effect, but after August the effects are minimal. The reverse is true, but with slightly less dramatic changes, when we return back to 2019 border traffic. Here the increased travels during April have a larger effect than reducing the traffic would have had, presumably due to the real traffic already being low during April.

The results for Denmark are similar to those of Finland and Norway during April—October 2020, but with less dramatic changes. During the fall, Denmark would have slightly suffered from removing the border traffic according to our model. Sweden stands out from the other Nordic countries, because the border traffic has very little effect on its epidemic situation.

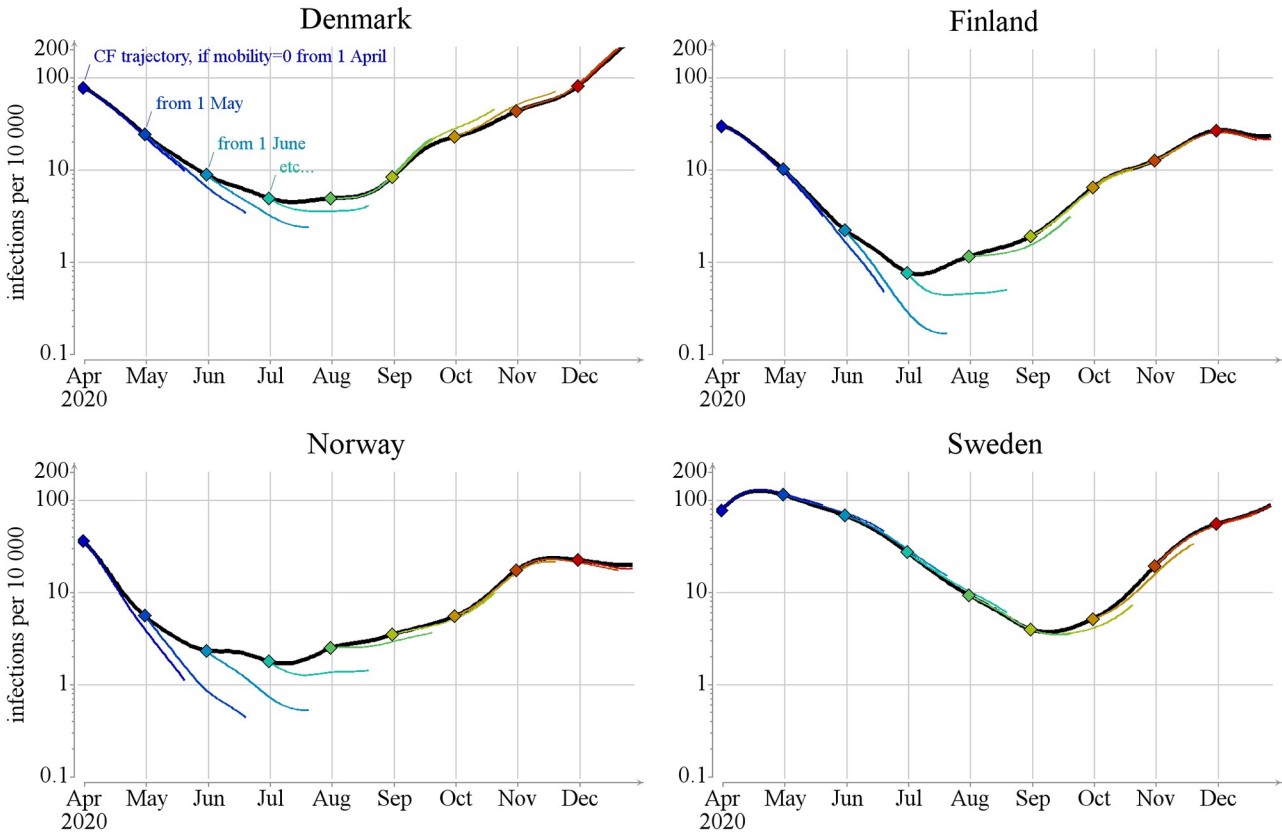

**Fig 8. Comparison to counterfactual scenarios for the border closure.** Black line shows the mean number of infections $E(I_{t,x})$ in the baseline scenario. Colored lines presents the number of infections in counterfactual scenarios, during the 50 day interval starting with the implementation of restrictions. Dots mark the start of the restriction.

Reduced border traffic in the fall, and increased border traffic during the spring and summer as this would have only slightly reduced the epidemic pressure inside the country.

## 5 Summary and discussion

Due to the economic, legal and social complications related to any mobility restrictions it is crucial to understand the efficacy of such measures during pandemic spread. To this end, we have undertaken an in-depth modeling study of the effect of border closures in the four neighboring Nordic countries during 2020. We used two kinds of metrics to estimate the effect of the mobility: descriptive and counterfactual. Descriptive metrics, like the net mobility effect, focus on estimating the direct consequences, i.e., numbers of infections arriving or leaving the country. Such metrics can help decision makers to focus on the most important factors controlling the spread and growth of the infections. From our study we can conclude that for inter-country mobility to have an effect on the spread of the infections, three conditions have to be met: (1) a low number of new infections in the target country, (2) a high number of infections in the exporter country, and (3) a sufficiently high level of mobility between the countries to transfer infection.

An interesting detail that emerges from our study is that, in our model, commuters had a consistently low contribution to disease transmission. In our model each commuter spent only half a day in the destination country per border crossing, while long-term travelers spend

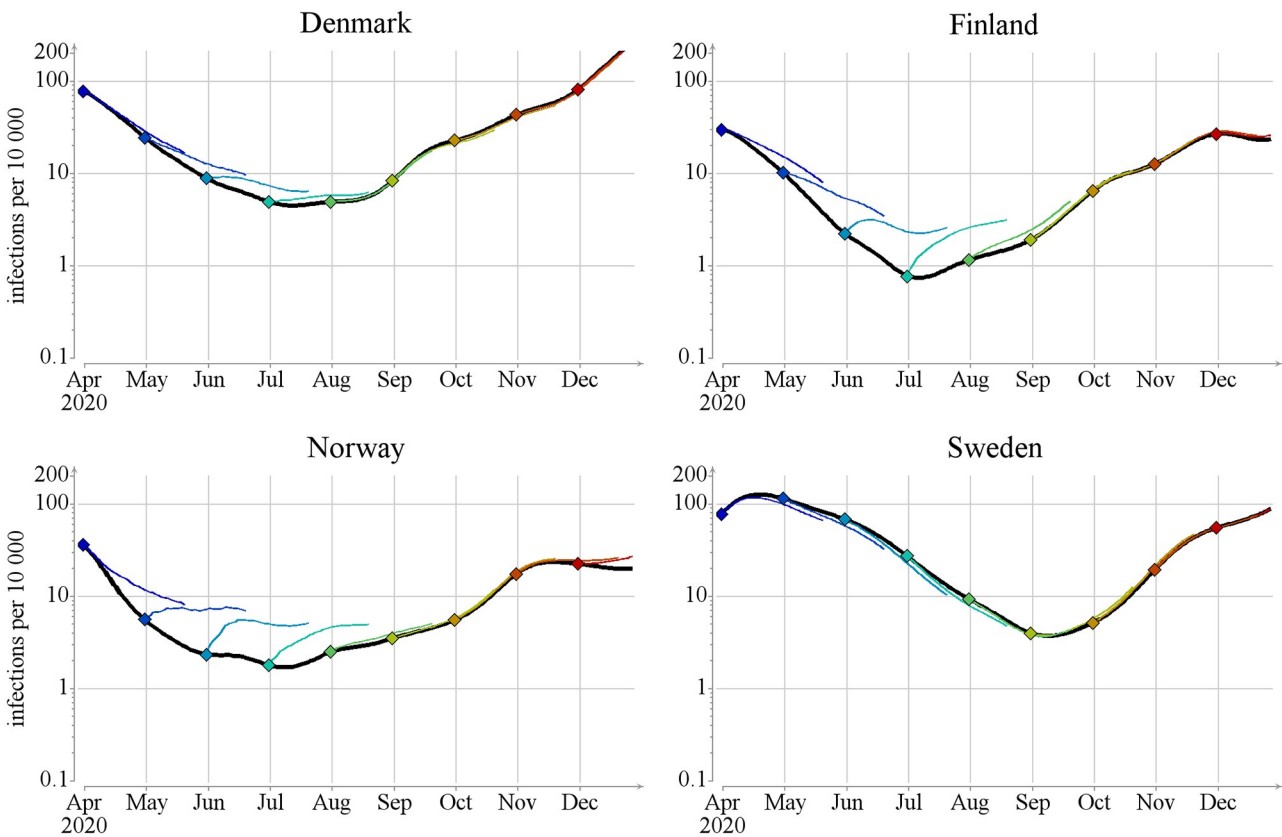

**Fig 9. Comparison to counterfactual scenarios for the border reopening.** Black line shows the mean number of infections $E(I_{t,x})$ in the baseline scenario. Colored lines presents the number of infections in counterfactual scenarios, during the 50 day interval starting with the implementation of border reopenings. Dots mark the start of the reopenings.

their whole infectious period (8 days on average); this creates a ratio of 1 to 16, which is further amplified by the low number of identified commuters. This conclusion relies on assumptions about the behaviour of long-term travellers—in reality they could spend much less time in the destination country. There is however a lack of reliable data of this type on long-term travellers.

We should also mention that there is a multitude of possible secondary effects that we have not included in our modeling: arriving infections may cause an outbreak, the outbreak may cause tightening of the restrictions which in turn may lead to reduction of local transmission, which in turn can delay the herd immunity effects etc. While these secondary effects are often negligible in studies estimating the time until the outbreak starts, our focus on investigating the effect of inter-country mobility during the outbreak necessitates their consideration. To this end, we have included two counterfactual scenarios that should constitute bounds for the effects of mobility restrictions, at least in the medium-short time period.

Our rather simplified model does not include stratification by age, region, or detailed disease states. This may prevent our results from being quantitatively exact, but we believe that the qualitative properties would not be affected. It should be added that, during the research project, it was found that detailed data about inter-country mobility is scarce. For instance, the rate of vehicles crossing a border point may be known, but not the number of individuals, or the time they spend in the receiving country. More detailed data on such aspects could improve the modeling accuracy.

There are other important differences between the real epidemic and an SIR model. Super-spreading is suggested as a driving factor for the pandemic during the studied period [27]. Super-spreading implies a heavy tailed distribution of secondary infections, while the deterministic SIR model assumes a constant number. In addition, assortative mixing has been suggested as a factor influencing the infection dynamics [28]. In our model we use a time-varying reproduction number $\mathcal{R}_t$, which should be able to capture the fluctuations caused by super-spreading and assortative mixing. It should also aggregate the differences between countries, e.g. differences in climate and pandemic suppression policies.

Leaving one country to visit another may change the number, duration and context of social contacts. In our model, we assumed that people crossing the border have the same infectivity and susceptibility as the local people in the visited country (including those doing local travel). Collecting quantitative data on travellers' behaviour abroad and estimating additional risks imposed by the travel itself would help in further planning of intervention measures.

National borders often constitute an ideal location for interventions due to a limited number of crossing points, and the existing infrastructure for controlling travel. Further, limiting movement within a country can be practically and legally more difficult than denying access to a country. The interventions can be achieved either via limiting the number of travellers or controlling and testing for symptomatic or asymptomatic passengers. However, assessing the effectiveness of inter-country travel restrictions can be more difficult than within-country travel, because one needs to combine data from multiple countries' health officials. In addition, within-country mobility is much better studied than inter-country mobility. Despite these difficulties, we believe that our modelling study benefits both the assessment of intervention strategies at the borders and modelling epidemic spread within a given country by separating the effect of the external disease pressure and thus yielding more accurate reproduction numbers.

## Supporting information

**S1 Appendix. Mobility data and model.**
(PDF)

**S2 Appendix. Derivation of the transmission model.**
(PDF)

**S3 Appendix. Additional results.**
(PDF)

## Author Contributions

**Conceptualization:** Mikhail Shubin, Hilde Kjelgaard Brustad, Felix Günther, Tapio Ala-Nissila, Gianpaolo Scalia Tomba, Mikko Kivelä, Lasse Leskelä.

**Data curation:** Hilde Kjelgaard Brustad, Jørgen Eriksson Midtbø, Felix Günther, Laura Alessandretti.

**Formal analysis:** Mikhail Shubin.

**Funding acquisition:** Tapio Ala-Nissila, Mikko Kivelä, Lasse Leskelä.

**Investigation:** Mikhail Shubin, Gianpaolo Scalia Tomba, Mikko Kivelä, Lasse Leskelä.

**Methodology:** Mikhail Shubin, Felix Günther, Tapio Ala-Nissila, Gianpaolo Scalia Tomba, Mikko Kivelä, Lasse Leskelä.

**Project administration:** Tapio Ala-Nissila, Mikko Kivelä, Lasse Leskelä.

**Resources:** Tapio Ala-Nissila, Mikko Kivelä, Lasse Leskelä.

**Software:** Mikhail Shubin, Laura Alessandretti.

**Supervision:** Mikko Kivelä.

**Validation:** Mikhail Shubin.

**Visualization:** Hilde Kjelgaard Brustad.

**Writing – original draft:** Mikhail Shubin, Hilde Kjelgaard Brustad, Felix Günther, Laura Alessandretti, Tapio Ala-Nissila, Gianpaolo Scalia Tomba, Mikko Kivelä, Louis Yat Hin Chan, Lasse Leskelä.

**Writing – review & editing:** Mikhail Shubin, Hilde Kjelgaard Brustad, Laura Alessandretti, Tapio Ala-Nissila, Gianpaolo Scalia Tomba, Mikko Kivelä, Louis Yat Hin Chan, Lasse Leskelä.

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
