## [Decision Letter · Decision Letter 0]

6 Feb 2024

Dear Dr. Shubin,

Thank you very much for submitting your manuscript "The influence of cross-border mobility on the COVID-19 epidemic in Nordic countries" for consideration at PLOS Computational Biology.

As with all papers reviewed by the journal, your manuscript was reviewed by members of the editorial board and by several independent reviewers. In light of the reviews (below this email), we would like to invite the resubmission of a significantly-revised version that takes into account the reviewers' comments.

All reviews emphasize the need for a clear and thorough discussion of the work's limitations. This discussion must assess the robustness of the conclusions in light of those limitations, especially regarding their consequences for decision-making.

We cannot make any decision about publication until we have seen the revised manuscript and your response to the reviewers' comments. Your revised manuscript is also likely to be sent to reviewers for further evaluation.

Sincerely,

Claudio José Struchiner, M.D., Sc.D.

Academic Editor

PLOS Computational Biology

Virginia Pitzer

Section Editor

PLOS Computational Biology

All reviews emphasize the need for a clear and thorough discussion of the work's limitations. This discussion must assess the robustness of the conclusions in light of those limitations, especially regarding their consequences for decision-making.

Reviewer's Responses to Questions

**Comments to the Authors:**

Reviewer #1: The manuscript: “The influence of cross-border mobility on the COVID-19

epidemic in Nordic countries” seeks to model the impact of between country border crossings on the spread of COVID-19 during the first year of the SARS-CoV-2 pandemic in 2020.

The authors use a rich set of border-crossing data available from four of the Nordic countries (Denmark, Norway, Sweden and Finland) to inform SIR models. These models are subsequently used to explore counterfactual scenarios relating to effects on the pandemic as a consequence of open or closed borders. Their findings suggests that border-crossing policies can have considerable influence on the evolution of a pandemic, more importantly, they find that there are certain instances where the influence of border-crossings on an epidemic are negligible. These findings are important for policy-makers and modelers alike to take into account when the next pandemic strikes. Moreover, there are only a few studies examining the effects of border crossings on pandemics such as COVID-19. Thanks to their cultural and geographic similarity, the Nordic countries provide excellent case studies that can potentially tease apart effects that can be relevant for many other western countries.

In my opinion, this is a very timely and interesting study examining cross-border effects on outbreaks, something that is lacking from the Nordic countries and therefore of great interest. By choosing four similar countries in terms of governance, geography and, to some extent, demography it is possible to control for the impact of country-specific measures on outbreaks. The manuscript is very well written and the model is thoroughly explained and appears to be sound. While there are limits as to what can go into a mathematical model of an infectious disease pandemic, there are some points I hope the authors could clarify and discuss to a greater extent than what has currently been done.

***Major revisions***

During the first year of the COVID-19 pandemic, there were many indications that super-spreading events were to some extent driving the pandemic (see 33139561[pmid]). How is this reflected by the model (if at all) and what impact can super-spreading events have on travel between countries? (i.e. can travel by boats, trains or planes function as super-spreading events? (see 35779143[pmid], 32726405[pmid]) How can such super-spreading events go on to drive secondary infections in the host countries? (This is also discussed somewhat in the referenced Creswell et al. article). It is not expected that the authors redo their analyses of course but a couple of sentences describing super-spreading and how this might be challenging to account for with standard SIR models would be helpful.

The ability to identify the true effect of boarder crossings on infectiousness is likely very difficult to assess as it is clearly challenging to separate local infections from imported. The values presented are therefore only projections assuming nothing but the reproductive number from the source country it seems. This should be highlighted as a premise for the study and reflected upon when the consequences and conclusions are presented, particularly for the counterfactual part. Traveling itself may introduce unidentified risks not accounted for and may not have been controlled for in the study. This should be pointed out in the discussion and recommendations section.

All countries considered are well-governed and although similar there are also some marked differences. While Finland and Norway are fairly similar in terms of both population size/density and geography (both countries also share borders with Russia), Sweden has a population almost double the size and thus lower population density. Denmark, on the other hand, is located on the European continent, away from Norway and Finland, and has an island-like structure with a markedly higher population density. As an EU member (as both Sweden and Finland also are), Denmark shares open borders with Germany in addition to Sweden. I miss more discussion on how these differences can impact the spread of COVID-19 internally in each country as a consequence of importations (See for instance 35229003[pmid]. It doesn’t have to be much but it should be mentioned how these differences are reflected in their model. The model is the same for all countries but the countries inform the model differently from the available data, can something have been lost?

What effect on the population does the length of immunity and the reproduction number in a country have on imported cases and their role in driving (or reducing) the pandemic? That is, if a fraction of the population is immune and immunity lasts for a long time (say 6-12 months) how does this influence the effects of border crossings on the evolution of the pandemic within a country? How can this differ from say shorter duration of immunity as witnessed by the more recent SARS-CoV-2 Omega strains? What about a country with a higher population density like Denmark, how might imported infections spread there compared to countries with a lower population density like Norway and/or Finland? Is it at all possible to separate the effects from importation in a country like Sweden with high infection rates from a country like Denmark with higher density, or will these countries appear the same from the proposed model?

Not much is revealed regarding the impact of Russian cross-border traffic in Norway and Finland in the main text, although some results are presented in the S1 appendix. How could these border crossings affect spread of COVID-19 and is it expected to differ from within Nordic countries border crossings?

***Minor revisions***

Is it Shubin Mikhail or Mikhail Shubin (see S1-S3 appendix authors).

In S1 appendix, 2. page 2. paragraph: make sure you indicate the “o” index in D_{t,d ←o} ^{long/short}

In S1 appendix, 2. last paragraph page 2: The superscripts ? (plural)

In S1 appendix, first paragraph, page 7: Note that the because…

It is not always completely apparent what the figures in S3 appendix designate, some more information in the figure legends would be appreciated.

***Remarks that the authors can decide whether have merit***

The initial reproduction number is defined to be Normal with mean=2 and sd=0.5, this can some times be negative, which is nonsensical, why not use log-normal?

In equation 13 the letters of the different terms end up with the same letter “I” that also designates infectiousness.

Reviewer #2: This is an interesting paper that uses a tractable model to investigate the impact of public health interventions, people movement in particular, on the spread of SARS-CoV2 in the Nordic countries. I find this work interesting and relevant. The results seem sensible, but they are very difficult to evaluate in absence of some key details that I list below, and which I consider essential before I am able to recommend revisions.

- Section 3.1.2 requires a more detailed explanation of how M is estimated. I suggest spelling out each term in equations 16 and 17.

- Similar to above, the final sentence in section 3.1.3, ‘Prevalence of infection among long-term travellers from outside Nordic countries is provided in the data’, is not entirely clear. Is this part of the data that was fed to the model and is it thus fixed? This is not the impression I got from reading the rest of the methods, but I could be entirely wrong.

- I appreciate the graphical depiction of the model (Fig 2). However, it would be very helpful to have an expression for what the likelihood function is here. There are many parameters and if the data are just the number of hospitalisations per day, then stating out the likelihood here would help put everything into context.

- In Fig 2 I can see that there are several fixed components in the model. Can the authors comment on their uncertainty? For example, whether the hospitalisation probability or its probability after t days of infection have associated uncertainties. In that case, what would be the effect of treating these as stochastic nodes? Presumably not much, but I think this point would help the reader understand the extent to which the model is realistic.

- The authors have commented on variations of their model due to age and regional stratification. However, there are other important factors, like those that break the SIR assumptions. I suggest adding a few sentences on whether these can have an impact here.

- There is substantial evidence that Sweden had the highest viral genetic diversity of the virus among Nordic countries in the early stages of the pandemic. This indicates that it acted as a net exporter of the virus. Of course, these dynamics probably changed rapidly, with local transmission playing an increasingly big role in each jurisdiction. In the light of this work, this can simply mean that the number of importations was not as important as the fact that they actually resulted in ongoing transmission in each country. Can the authors comment on how the genetic evidence stacks up and its implications?

- There are a few minor typological or grammatical errors that can be easily fixed with careful proofreading. E.g. line 243 ‘between by country’ – remove the ‘by’.

Reviewer #3: As an epidemiologist reviewer, my primary focus is on the results and reporting. The paper titled 'The Influence of Cross-Border Mobility on the COVID-19 Epidemic in Nordic Countries,' authored by Dr. Mikhail Shubin and colleagues, presents an elegant study. The authors have quantitatively estimated the impact of cross-border traffic on the number of COVID-19 infections in Nordic countries during the first year of the pandemic. The study features a clear and concise presentation of the conditions necessary for inter-country mobility to influence the spread of infections. The listed criteria, including (i) very low new infections in the target country, (ii) a significant disparity in infections between the source and target countries, and (iii) a substantial level of mobility, provide a robust framework for understanding when inter-country mobility becomes a critical factor in infection transmission. The authors also emphasize the increase in imported infections in Nordic countries during the summer of 2020, attributed to infected travelers from Sweden and, later in 2022, from Denmark. This information is concisely described and offers valuable insights into the source of these infections.

Furthermore, the authors present a particularly noteworthy scenario in which Meff > 1 and Reff < 1, indicating that the epidemic could propagate solely through inter-country mobility. This specific situation was observed in Finland and Norway during the summer of 2020. Additionally, substantial differences in prevalence between local and traveling populations are noted, with Nordic travelers being more likely to be infectious in Norway and Finland. In contrast, the opposite effect is observed in Sweden. The authors also explore counterfactual scenarios for border closures and reopenings.

In summary, this article makes a valuable contribution to our understanding of the impact of international mobility on the COVID-19 pandemic, offering a comprehensive analysis of the specific case of the Nordic countries. The manuscript is suitable for publication in PLOS Computational Biology.

Suggestions for improvement:

1. One aspect that the article does not address is the distinction between different COVID-19 virus variants. While the dominance of the Wuhan variants is evident for most of 2020, the emergence of the VOC Alpha variant in December is not sufficiently highlighted.

2. Furthermore, enhancing the work by including comparisons to other models or data from different countries could enrich the discussion and conclusions.

Reviewer #4: Shubin et al. explore the effects of cross-border movement on the trajectory of the early SARS-CoV-2 pandemic in Nordic countries. They employ data-driven modelling of concurrent infection trajectories in each country, including parameters for short and long term travel between each and the rest of the world. They also consider counterfactual scenarios to corroborate their results. Overall, they show that the effects of travel can be significant, but only if particular conditions are met. These results add nuance to the broader discussion on border restrictions during the pandemic.

I want to commend the authors for undertaking this work given the difficulty in collating the data here. I would like to see it published following revision. While the science appears sound, the manuscript is lacking detail or unclear in places. My comments largely pertain to making it clearer.

Major Comments:

L115-121: The text around Gaussian mixture models here is quite confusing. It would b

---

## [Decision Letter · Decision Letter 1]

20 May 2024

Dear Dr. Shubin,

We are pleased to inform you that your manuscript 'The influence of cross-border mobility on the COVID-19 epidemic in Nordic countries' has been provisionally accepted for publication in PLOS Computational Biology.

Also note that Reviewer 4 has some minor suggestions that can be addressed at the proof stage prior to publication.

Best regards,

Claudio José Struchiner

Academic Editor

PLOS Computational Biology

Virginia Pitzer

Section Editor

PLOS Computational Biology

Please note that Reviewer 4 has some minor suggestions that can be addressed at the proof stage prior to publication.

Reviewer's Responses to Questions

**Comments to the Authors:**

Reviewer #1: My comments have been answered.

Reviewer #2: I would like to thank the authors for their effort in improving the clarity. They have addressed many technical comments from four reviewers, which implied substantial editing and careful responses.

I am satisfied with the work they have done, and particularly their explanation of parameter identifiability, the definition of the likelihood functions and the potential impact of violations to this SIR model (e.g. superspreading). I do not have additional comments and am happy to recommend this paper for publication.

Reviewer #3: The authors meticulously addressed all my concerns. After careful consideration, I find no further points to critique. I commend the authors for their diligent efforts and thoroughness of work. Given the quality and significance of their findings, their manuscript deserves publication in the PLOS CB journal. Congratulations to the authors on their excellent contribution to the field.

Reviewer #4: The authors have addressed all of my comments. I think the manuscript has improved and I do not need to see it again.

It will be nice to see an epidemiological study looking at the Nordic countries in addition to the genomic work in the literature, and this paper unifies the results well.

I only have a couple of minor corrections to make, and urge the authors to check carefully for typos at the proofing stage, since the expression is sloppy in parts.

Minor:

L5: economical

L96: Stick to SARS-CoV-2 as this is the virus, like you have listed for other viruses like Ebola, SARS, and so on. Covid-19 is the name of the corresponding disease. I am sorry to be pedantic, but this is an important distinction to be consistent with the rest of the literature.

L104: Millions → million

L281: “exporter of mobility” doesn’t make sense. Maybe ‘exporter of infections’?

Fig3: I would suggest a line pointing out that negative values correspond to the export of infections. This would make things much clearer, since I found I needed to reference the figures before parsing the text.

L415: “Benefited” -> “exported”, as per my comment in initial reviews.

**Have the authors made all data and (if applicable) computational code underlying the findings in their manuscript fully available?**

Reviewer #1: Yes

Reviewer #2: Yes

Reviewer #3: Yes

Reviewer #4: Yes

PLOS authors have the option to publish the peer review history of their article (what does this mean?). If published, this will include your full peer review and any attached files.

Reviewer #1: No

Reviewer #2: No

Reviewer #3: No

Reviewer #4: No

---

## [Editor Report · Acceptance letter]

10 Jun 2024

PCOMPBIOL-D-23-01783R1 

The influence of cross-border mobility on the COVID-19 epidemic in Nordic countries

Dear Dr Shubin,

I am pleased to inform you that your manuscript has been formally accepted for publication in PLOS Computational Biology. Your manuscript is now with our production department and you will be notified of the publication date in due course.

With kind regards,

Zsofia Freund
